# Effect of Pure Dephasing Quantum Noise in the Quantum Search Algorithm Using Atos Quantum Assembly

**DOI:** 10.3390/e26080668

**Published:** 2024-08-06

**Authors:** Maria Heloísa Fraga da Silva, Gleydson Fernandes de Jesus, Clebson Cruz

**Affiliations:** 1Grupo de Informação Quântica e Física Estatística, Centro de Ciências Exatas e das Tecnologias, Universidade Federal do Oeste da Bahia—Campus Reitor Edgard Santos, Rua Bertioga, 892, Morada Nobre I, Barreiras 47810-059, BA, Brazil; maria.fraga@fbter.org.br; 2Latin American Quantum Computing Center, High Performance Computing Center, SENAI CIMATEC, Av. Orlando Gomes, 1845, Piatã, Salvador 41650-010, BA, Brazil; gleydson.jesus@fieb.org.br

**Keywords:** quantum computing, Grover’s algorithm, software development, AQASM, quantum noise

## Abstract

Quantum computing is tipped to lead the future of global technological progress. However, the obstacles related to quantum software development are an actual challenge to overcome. In this scenario, this work presents an implementation of the quantum search algorithm in Atos Quantum Assembly Language (AQASM) using the quantum software stack my Quantum Learning Machine (myQLM) and the programming development platform Quantum Learning Machine (QLM). We present the creation of a virtual quantum processor whose configurable architecture allows the analysis of induced quantum noise effects on the quantum algorithms. The codes are available throughout the manuscript so that readers can replicate them and apply the methods discussed in this article to solve their own quantum computing projects. The presented results are consistent with theoretical predictions and demonstrate that AQASM and QLM are powerful tools for building, implementing, and simulating quantum hardware.

## 1. Introduction

The advent of quantum computing is one of the bases of the so-called second quantum revolution [1,2,3,4]. The scientific community ceased to be mere observers of the laws of quantum mechanics and is now actively acting to use them in the development of practical quantum devices [1,2]. In this scenario, developing devices that work according to the laws of quantum mechanics is one of the most ambitious goals of the current century. This revolution is expected to be responsible for the major technological breakthroughs of the 21st century, and millions of dollars are already being invested in research into the development of quantum hardware by companies and universities around the world [5,6]. The advent of this technological revolution also marks the onset of the Noisy Intermediate-Scale Quantum (NISQ) era [7,8,9], characterized by the development of quantum devices with a limited number of qubits and high susceptibility to noise [8]. In this scenario, the NISQ era provides an enticing insight into the capacity of quantum computing to revolutionize fields ranging from cryptography and optimization to drug discovery and materials science [10].

Nevertheless, the realization of this potential depends on successfully overcoming challenging technological obstacles [8]. NISQ devices are inherently prone to errors arising from decoherence, gate imperfections, and environmental noise [11,12]. Moreover, developing fault-tolerant quantum error correction codes is challenging and requires new methods to reduce the effects of noise on quantum computing [9].

In this context, our work delves into the intricacies of quantum noise and its impact on quantum algorithms. We present a study on the effect of pure dephasing quantum noise in the quantum search algorithm, employing an implementation of the 4-qubit quantum search algorithm on the Atos Quantum Assembly Language (AQASM) [13,14]. Leveraging the my Quantum Learning Machine (myQLM) quantum software stack [15,16], we provide a practical framework for writing and executing quantum algorithms through a Python interface. Our approach enables researchers to conduct noise-free simulations using the PyLinalg linear algebra simulator via myQLM, all from the convenience of a standard home computer. Importantly, we make our programming code readily available throughout the text, facilitating reproducibility and enabling others to build upon our work.

However, the implementation of Grover’s algorithm, a cornerstone of quantum search, presents challenges due to the necessity for multi-controlled quantum gates. To address this obstacle, we utilize the NoisyQProc simulator, an invaluable resource within the Quantum Learning Machine (QLM) programming development platform [17]. While this simulator is a private version of myQLM and not publicly accessible, it allows for the emulation of real quantum processors with adjustable architectures, enabling comprehensive analysis of quantum noise effects in the quantum search problem.

The advantages of myQLM include open-source interoperability connectors with different open-source frameworks like Qiskit [18], Cirq [19], ProjectQ [20], and Forest™ [21], and users can also create additional connectors themselves. Through myQLM’s plug-in architecture, users have the option to develop their hardware architecture with personalized qubit connections and develop their own pre- and post-processing techniques to enhance the efficiency of their quantum circuits with respect to a particular target quantum technology. Consequently, they can benefit from enhanced performance, expanded simulation capabilities, and advanced features like quantum circuit optimizers and noise simulators. Overall, myQLM provides users with a versatile platform to explore and experiment with various quantum computing technologies, ultimately enabling them to push the boundaries of quantum computing research and development. This flexibility and customization make myQLM a valuable tool for both beginners and experienced researchers in the field of quantum computing.

Recent studies [22,23] mapped quantum noise in Grover’s algorithm on real quantum computers as a function of the number of qubits used, demonstrating the scalability issues of the algorithm on real quantum computers. However, these studies do not provide a pathway to determine the hardware requirements necessary for the algorithm’s scalability. At this point in quantum computing, real processor simulations are constrained by the hardware used, preventing the extrapolation needed to assess the algorithm’s quality based solely on noise levels. These tests also preclude variations in the application time of logic gates and the architectures used. Conversely, simulations allow for the assessment of topologies, noise levels, and ideal gate application times in real quantum computers for executing specific quantum algorithms.

In this context, our results focus on presenting the impact of different relaxation times on the performance of Grover’s algorithm within an optimal architecture simulated using myQLM. By evaluating pure dephasing quantum noise, we highlight the importance of considering relaxation times when designing optimal architectures for quantum algorithms. Our findings underscore the efficacy of AQASM as a versatile tool for building, implementing, and simulating quantum hardware, thus bridging the gap between theory and practice in the NISQ era. By elucidating the challenges and opportunities inherent in quantum computing, we contribute to the ongoing dialogue surrounding the development of practical quantum technologies and pave the way for future advancements in the field.

## 2. Quantum Search Algorithm

Considering the lack of myQLM papers in the literature, this section is didactically tailored for readers with low experience in quantum computing. First, let us start with an overview of the four-qubit quantum search algorithm. The search problem is a very common topic in classical computing. Considering an unstructured database with *N* entries, the problem consists of determining the index of the database entry (*x*) that satisfies some predefined search criteria x=y, where *y* is the searched element—a brief explanation of the algorithm can be found on supplementary material [24].

Assuming Grover’s algorithm for n=4 qubits, one can create a list of N=16 items, represented by each state of the computational basis of a 4-qubit system. Table A1 in Appendix A shows the list of 16 items for which we chose color names randomly arranged. Each item is associated with a number from 0 to 15, as listed in the second column. Finally, as Grover’s algorithm performs the search on the state vectors, each binary entry is associated with a state vector in the computational basis for 4 qubits, as per the last column.

In the following, we present the four steps that characterize Grover’s algorithm: initialization, Oracle, amplitude amplification, and measurement [25], along with the integral code written using AQASM language and myQLM quantum software stack.

First of all, we recommend installing the Jupyter Notebook environment, which is available for free. In order to do this, we recommend installing the free Anaconda platform, which can be downloaded for Windows, macOS and Linux operating systems from the official website [26]. Once installed, the platform’s interface on the home page displays applications including Jupyter Notebook. By clicking on “launch”, the user is automatically directed to the Jupyter environment, where it is possible to create a Python-compatible ipynb file and perform actions explained in the Jupyter documentation [27]. Secondly, the myQLM tool must be installed and it is also available for free using the official guide provided [28]. Once the above installations have been made, one can proceed to write and run the code shown in the boxes below using AQASM, which documentation is available on the official GitHub [13].

Finally, both the Jupyter Notebook file in myQLM regarding the boxes in Section 2 for the noise-free simulation and the Jupyter Notebook file in QLM regarding the Boxes in Section 3 for the noisy simulation are available on a GitHub repository created by the authors for easy access by readers [24].

Once the integrated development environment has been established, the first step is to import the computational tools needed to implement the quantum algorithm in AQASM. Box 1 shows the command cell that imports the computational tools required to implement Grover’s algorithm in AQASM.

Box 1Importing the computational tools.

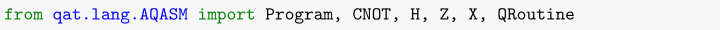



Subsequently, we allocate the amount of quantum and classical bits that will compose our quantum circuit, as shown in Box 2. The same quantity is defined for both since the bits will store the results of the measurements performed on the qubits to identify their final states [25].

Box 2Allocating qubits and classical bits registers.

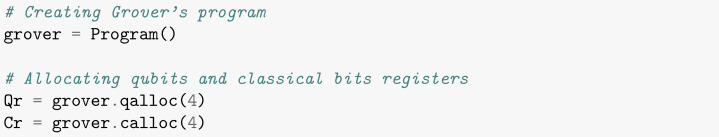



### 2.1. Initialization

In order to initialize the qubits in a balanced superposition represented by the state |S〉, one can apply the Hadamard gate on all the qubits
(1)$$|S\rangle =H^{⊗4}|0000\rangle .$$

In terms of coding, Box 3 shows the initialization process of Grover’s algorithm.

Box 3Initializing the qubits in a balanced superposition.

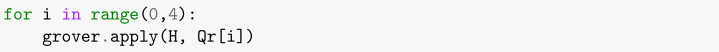



### 2.2. Oracle

In the following, we must build the Oracle that searches for the desired item. There are two main methods used in the literature to build the Oracle subroutine: the boolean and phase inversion methods [25,29,30]. In the boolean technique, the presence of an auxiliary qubit, also known as ancilla, initialized in the |1〉 state, is necessary. In this scenario, the ancilla is changed only if the input to the circuit is the sought state. However, this method is analogous to the classical search problem [29,30] and is generally applied to compare the computing power of the quantum superposition principle for quantum computing [30].

Thus, for the purposes of simplicity, we opted for the most straightforward method, the phase inversion method [29,30]. This method excludes the need for an ancilla, and the Oracle’s role in this process becomes to identify the sought element in the balanced superposition of the states of the computational base described above and to add to it a negative phase. In this context, the Oracle function can be represented by the unitary operator:(2)$$\begin{array}{c} O|x\rangle =\left\{\begin{array}{cc} -|x\rangle & \mathrm{se}\;x=y,\\ \;\;|x\rangle & \mathrm{se}\;x\neq y, \end{array}\right. \end{array}$$
where |x〉 is a state of the computational basis. Therefore, *O* can be defined as a diagonal matrix, which adds a negative phase to the state corresponding to the searched item |y〉. It is important to highlight that there is an Oracle to search for each state vector on the 4-qubit computational basis.

Here we arbitrarily choose to find the item **blue**, associated with decimal 15, which correspondent state vector is |1111〉 according to Table A1. Table A2 in Appendix B shows the quantum circuits for all Oracles in the 4-qubit Grover’s algorithm. It is important to clarify that, in order to simplify the quantum computing process, the Oracles shown here are toy models. When it comes to applying Grover’s algorithm to practical problems, the design of the quantum circuits for the Oracles would need to be readjusted [30].

The Oracle responsible for marking the item **blue** is composed solely of the multi-controlled gate Z (MCZ), having qubits 0 to 3 as the controls and qubit 4 as the target, as shown in Box 4.

Box 4Creating the Oracle subroutine that marks state |1111〉.

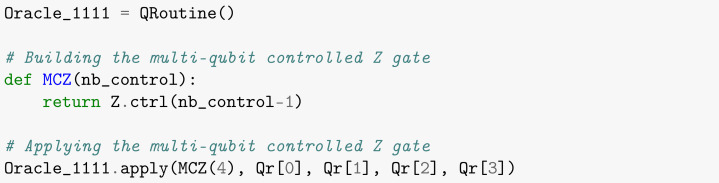



However, even having indicated the sought element with a negative phase, the Oracle routine is still insufficient to guarantee that the searched element in the list will be found if we measure our balanced superposition. This is due to the fact that adding the phase in the sought state does not affect the probability distribution of the global state. In this sense, the probability of finding the searched item is 1/N (6.25%), which is equivalent to the classical Oracle in a single query in the list. Therefore, it is necessary to amplify the probability of the sought element, increasing the chance of finding it and reducing the probabilities of the other states in the process. This step is performed by the amplitude amplification method [29,30,31].

### 2.3. Amplitude Amplification

In contrast to the Oracle, the amplitude amplification subroutine is the same, regardless of the state representing the item sought. The amplification is carried out by performing a reflection represented by the unitary operation
(3)A=2|S〉〈S|−I,
increasing the amplitude of the searched item |y〉 [30]. Box 5 shows the cell that creates the amplitude amplification subroutine. First, we apply the Hadamard gate to all qubits in the Oracle-modified state. Then, gate X is used to flip all qubits and we apply the MCZ gate. Finally, the process is finished by flipping all qubits again and applying the Hadamard gate, as shown in Box 5, obtaining the final state and amplifying the probability of the sought item being found.

Box 5Creating the Amplitude Amplification subroutine.

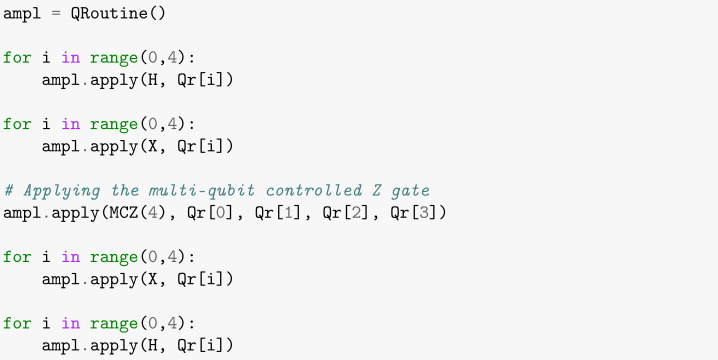



When we reach the final state, Grover’s algorithm is completed by measuring the amplified state. The probability of finding the searched item in a single measurement after the amplitude amplification process in the 4-qubit algorithmic probability is 39.0625%, which is an improvement compared to the 6.25%, achieved in the classical algorithm in a single query of the Oracle, but still insufficient to assure that the search will be successful. Thus, in order to achieve the full potential of Grover’s algorithm, subroutines Oracle and amplitude amplification need to be repeated to maximize the probability of finding the sought state in the measurement of the quantum state [29]. The algorithmic probability of finding the searched item after *r* repetitions is given by
(4)$$P={\left[\frac{2}{\sqrt{N}}\left(\frac{N-2r}{2N}+\frac{N-r}{N}\right)\right]}^{2}.$$

From Equation (Equation 4), it is possible to verify that $r=\sqrt{N}$ is enough to assure that the user obtains the sought state after the measurement. Figure 1 shows a sketch of the process, with a pictorial representation of the probability amplitudes in the 4-qubit scenario.

Since it is up to the users to inform which item they are looking for, it is possible to make available, at the beginning of the code, an interaction instructing them to type the name of the item so that our software applies the corresponding Oracle (Box 6). In this context, the conditional statement if was used for item 0, elif for items 1 to 15, and else if the user enters an item that does not belong to the list.

At this point, the user would type the input **blue** implying the application of the subroutines Oracle and amplitude amplification. The following output is displayed:


Which item do you want to find? Use initial capital letters. Blue


Box 6Asking for user input.

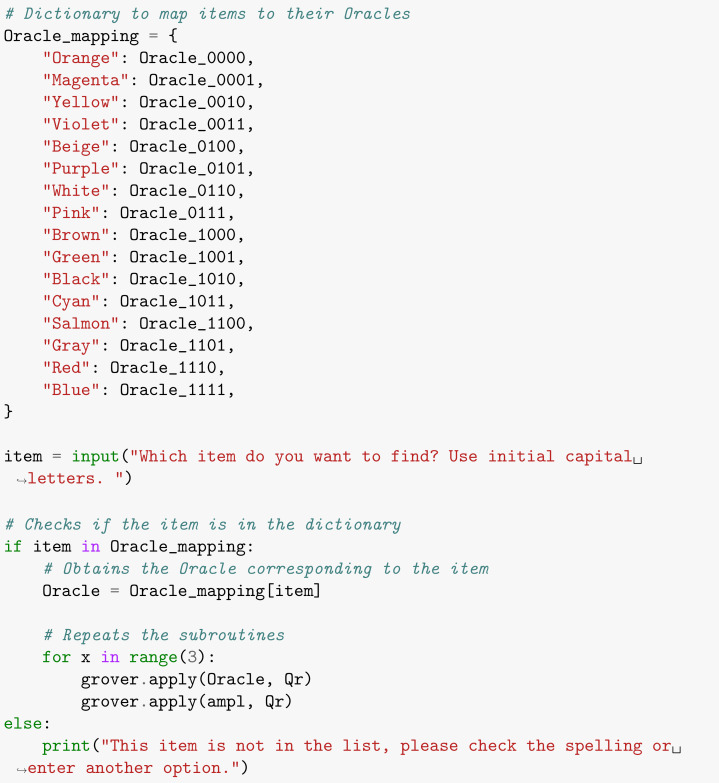



### 2.4. Measurement

The fourth and final step of the algorithm is to perform measurements. Here, we will store the final result of each qubit in the corresponding classical bit via the *measure* command, as illustrated in Box 7. Note that it is necessary to transform the program into a circuit before representing it graphically in a drawing, which is laid out in SVG format as in Figure 2.

Box 7Measurements.

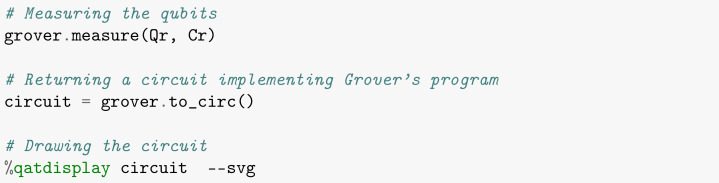



We can visualize below the complete circuit of Grover’s algorithm for the **blue** item search, assembled through the commands entered so far.

Therefore, using the PyLinalg package as a simple numpy-based simulator for quantum circuits [32] for data processing, one can turn the search results into histogram format, as shown in Figure 3. Moreover, PyLinalg simulates the execution of quantum circuits on a local processor and returns the counts of each measurement in the final state for a given set of repetitions (or shots) of that circuit in quantity defined by the user. Increasing the number of shots optimizes the exploratory simulation process, bringing it closer to the theoretically predicted result [25]. As shown in Box 8, we performed $2^{13}$ shots.

Box 8Simulating and plotting the results.

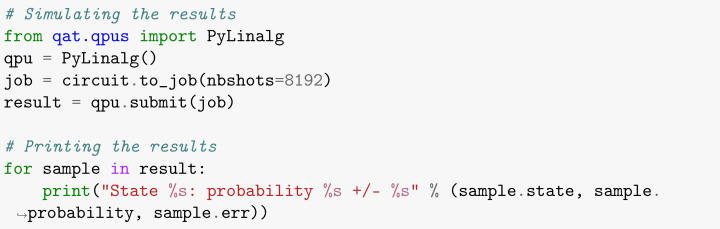



As can be seen, the searched item was found with 95.86% probability, while the other items in the list shown in Table A1 did not reach 0.5% probability individually. This significant increase in probability is related to the application of the repetitions of the Oracle and amplitude amplification subroutines, as shown in Equation (Equation 4). This is in contrast to the classical scenario since the probability of finding an item in an unstructured list with *N* = 16 entries, running just one query to the list, is 6.25%. This result highlights the benefit of using quantum features such as superposition to process information. In addition, while classically, the Oracle needs to query the list *N*/2 on average, the quantum algorithm can find the marked item in $\sqrt{N}$ attempts using Grover’s amplitude amplification method for solving the search problem. In conclusion, it is shown that the combination of Oracle and amplification subroutines for developing Grover’s algorithm results in a quadratic acceleration of the search problem, demonstrating that quantum computers have a major advantage over conventional computers.

## 3. Quantum Noise Interference on Grover’s Algorithm

Despite its clear advantage, Grover’s algorithm requires the application of multi-controlled quantum gates [25], which is an obstacle to implementing its 4-qubit and up versions in some quantum processors architectures presents in the popular IBM Quantum Experience [33] platform, for example, without a transpilation process. Consequently, there is a barrier to the scalability that would be required to make some algorithms useful and marketable on a large scale [8,34]. On the other hand, the number of qubits that are accessible in the system can be increased to substantially benefit the Grover algorithm. This would result in an exponential rise in the amount of storage space available in the database, which is the location where the searches are carried out. In this context, we carried out emulation of a genuine quantum processor by making use of myQLM as a means of simulating the interconnectivity amongst the necessary qubits in an effort to overcome this challenge of compatibility.

In addition, the existence of noise that affects the qubits during the implementation of quantum operations or even during idle time is the primary obstacle that prevents present quantum computers from reaching their full potential. In this scenario, Grover’s algorithm is one of the main hostages of noise since the quantum advantage of this algorithm can be better leveraged in quantum computers with a large number of qubits, which implies a large amount of noise and requires error correction protocols.

It is essential to note that all measurements conducted on a system are impacted by noise to varying degrees. This is anticipated by quantum physics, which postulates that quantum states collapse instantaneously at the point of measurement. This disturbance is inherent to the theory and cannot be prevented by any measuring technique [35]. Therefore, in order to simulate the quantum noise interference of the quantum processor on Grover’s algorithm, we use the quantum computing simulator CIMATEC KUATOMU, an ATOS QLM simulator, located at SENAI CIMATEC’s HPC center in Brazil, to build a simulated quantum hardware topology, since the AQASM noisy section is restricted to QLM users.

### 3.1. Quantum Hardware Model Simulation

A challenge faced by NISQ quantum computers is the direct application of multiple qubit gates, which requires quantum computers with high connectivity between the qubits and high coherence times [36,37]. However, this challenge can be overcome through a process of transpilation of quantum circuits, which can be used to reduce the number of controls in a logic gate [36]. Here, we have reduced four-qubit gates from (CCCZ) into one- and three-qubit gates (H and CCNOT), an approach that is easier to implement in real quantum devices. An advantage of the quantum simulator used is that it can be used to generate realistic topologies, serving as tests for the design of architectures with real connections and coherence times for the application of quantum algorithms that require specific topologies for the application of certain algorithms [38,39].

To simulate our quantum hardware model, we adjust our code to a maximum arity port of 3, which means that a quantum gate can be applied to a maximum of three qubits simultaneously. For this purpose, we decomposed the CCCZ gate into a set formed by Hadamard and CCNOT gates, as illustrated below. Consequently, it is necessary to introduce an auxiliary qubit for the circuit implementation to decompose this gate.

Therefore, to implement this new circuit, we created a specific 5-qubit topology in order to support the 4-qubit Grover algorithm and the auxiliary qubit of the decomposed CCCZ gate. All qubits are linked for the sake of simplicity; as a result, all of them can be concurrently controlled or targeted by controlled quantum operations. Figure 4 shows a sketch of the 5-qubit topology in which the 4-qubit Grover algorithm is implemented.

Similar to how we completed the noise-free simulation in Section 2, we needed to import the logic gates we would use, as seen in Box 9.

Box 9Importing the computational tools.

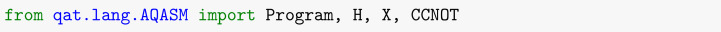



In this scenario, the allocation of classical bits and quantum bits takes place in the exact same manner as it did in the previous simulation. The fact that we employ one of these qubits as an ancilla necessitates that we now assign 4 classical bits and 5 quantum bits in our system (see Box 10).

Box 10Allocating qubits and classical bits registers.

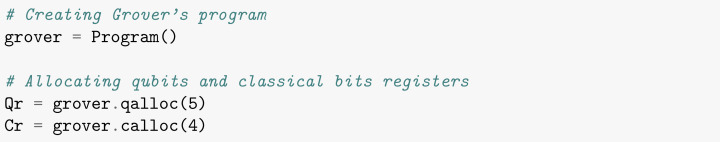



In order to build the topology connections for the quantum hardware (as shown in Figure 4), we must import the libraries Topology and HardwareSpecs. Box 11 contains the coding for the list items that represent the connections between the qubits in the idealized topology. The first member of each ordered pair is the control qubit, and the second element is the target one.

Box 11Defining the topology of the simulated quantum hardware.

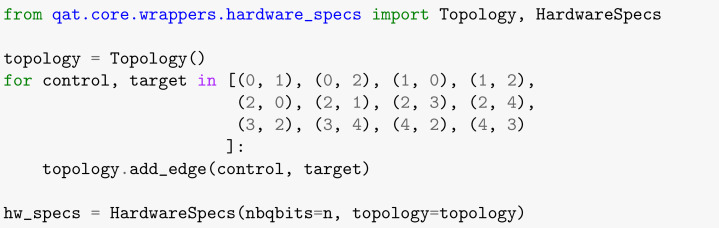



In Box 12, the initialization of the qubits occurs during the first step of Grover’s algorithm by putting them in a balanced superposition. Since qubit 2 is a supplementary qubit in this instance, it is unnecessary to start it in a superposition state.

Box 12Initializing the qubits in a balanced superposition.

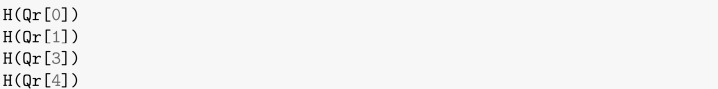



The second and third phases of the quantum search algorithm, implemented on the simulated quantum hardware are shown in Box 13. Initially, we apply the Oracle that explicitly looks for the state |1111〉 by applying the MCZ gate decomposed as illustrated in Figure 5. Next, we use amplitude amplification to enhance the probability that the marked object will be discovered while decreasing the probability that other items. These two procedures are repeated $\sqrt{N}$ times in order to maximize the probability of finding the sought state in the same way as performed in the noise-free simulation in Section 2.

Box 13Applying the Oracle and Amplitude Amplification subroutine that marks state |1111〉.

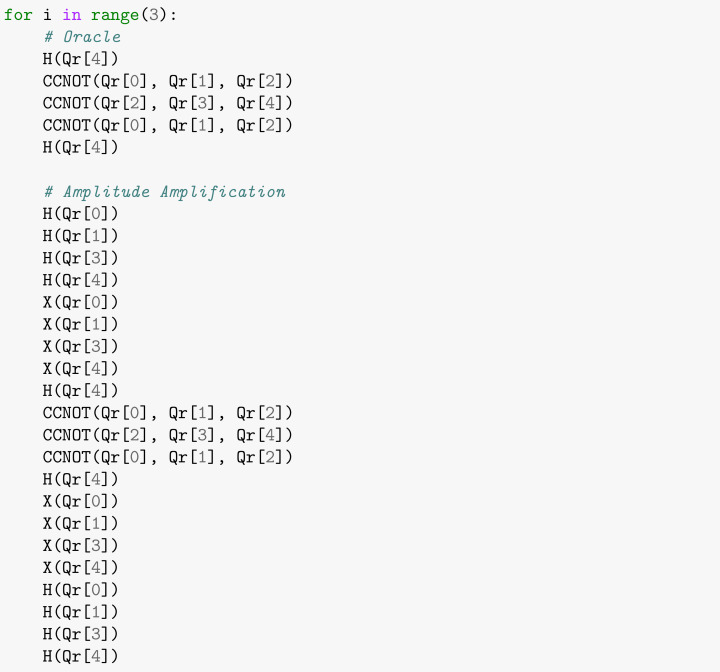



After this, we will finish the last step of Grover’s algorithm, which consists of taking measurements in accordance with Box 14. It is worth noting that, the execution of the measurement on the auxiliary qubit is completely unnecessary.

Box 14Measurements.

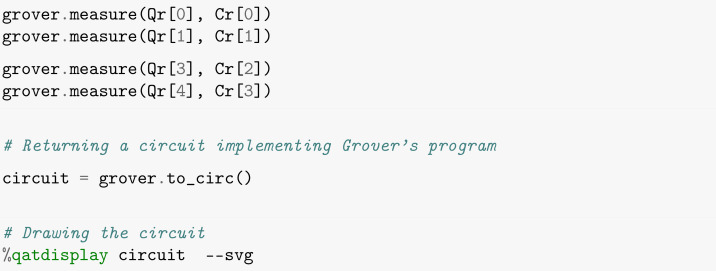



### 3.2. Quantum Noise

In the following, we introduce the quantum noise model in the simulated quantum hardware. Generally, to quantify quantum noise in quantum hardware setups, the measurement of two constants is performed: qubit relaxation time $T_{1}$ (i.e., *longitudinal relaxation* or *amplitude damping*) and qubit dephasing time $T_{2}$ (i.e., *transverse relaxation*) [40,41,42]. There is also a third parameter called pure dephasing time which is often the dominant contribution to $T_{2}$ [43,44]. This parameter determines how long a system will be able to keep its coherence, whereas amplitude damping offers a model for the physical process of energy decay associated with the presence of quantum noise [45] and it is represented by the following equation:(5)$$T_{\phi }=\frac{1}{\frac{1}{T_{2}}-\frac{1}{2T_{1}}}$$In this regard, we introduce the pure dephasing channel in order to perform the noisy simulations on the quantum processor emulated with the topology described in Figure 4.

In order to perform the noisy simulation, there are some properties that must be specified: the running time of the quantum gates used to build the quantum hardware topology and the operating time for the quantum gates used in building quantum hardware topology. The gate application and relaxation times are properties of each quantum computer, which can be altered once the calibrations are performed. Therefore, we selected these settings not with the objective of recreating the results of a particular system but rather with the only purpose of demonstrating the impact of these noises on the outcomes of the simulation. Consequently, it will be possible to make a comparison between the results of the noisy simulation and the results of the ideal noise-free simulation performed in the previous section.

Box 15 imports the library DefaultGatesSpecification and specifies the application timings of the gates used in Grover’s algorithm. Therefore, regarding the time needed to complete the procedures, we considered the following gate times in the code: X gate → 35.5 ns; Hadamard gate → 35.5 ns; CCNOT gate → 350 ns, measurements → 35.5 ns.

Box 15Defining gate times.

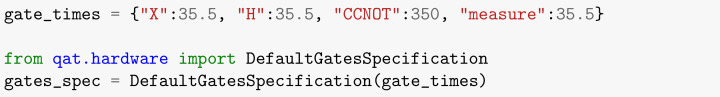



Furthermore, in order to apply the pure dephasing channel in Box 16, we need to import the library known as ParametricPureDephasing. In addition to the noise model, one can define the qubit relaxation times $T_{1}$ and $T_{2}$, respectively, as shown in the second code line of Box 16. Finally, in the last code line we use Equation (Equation 5) to define $T_{\phi }$.

Box 16Defining qubit relaxation times.

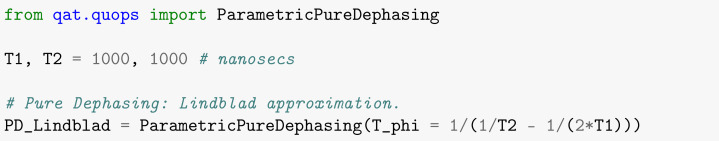



Subsequently, after importing the HardwareModel and NoisyQProc libraries, we specify the model of the quantum processor that we will use, including the gate specifications and the noise model that will be applied (see Box 17). Note that these parameters were defined by us in Box 15 and Box 16.

Box 17Importing quantum hardware.

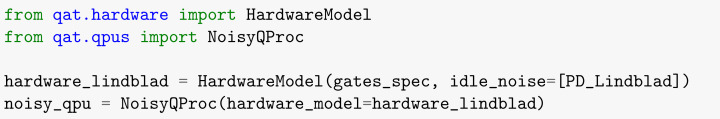



The last step of our code is to simulate Grover’s algorithm in the emulated quantum hardware under the pure dephasing noise model, as presented in Box 18. First, we need to create a job and link it to the quantum circuit created in Box 14 from Grover’s program developed in the previous steps, as in the first code line of the following box. The parameter nbshots indicates the number of repetitions used in the circuit, as will be explained later. Finally, these instructions were applied only to qubits 0, 1, 3 and 4 because, as explained before, qubit 2 is an auxiliary one.

Box 18Simulation.

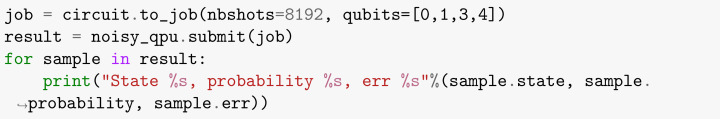



The simulations were conducted out using the stochastic technique, which requires that the density matrix be interpreted using a probability distribution based on pure states [29]. Using this method, the error scales with $\frac{1}{\sqrt{shots}}$. Thus, we perform the simulation with $2^{13}$ shots to obtain proper results.

Figure 6 shows the probability distributions of Grover’s algorithm performed in quantum hardware simulated under the pure dephasing noise model. Based on the results obtained from this analysis, we can gradually observe the effects of noise in the application of the algorithm by decreasing the dephasing time ($T_{2}$). It is worth noting that, although the most probable state is still the searched item |1111〉, the other states arose with significant probabilities even after the $\sqrt{N}$ repetitions of the Oracle and amplitude amplification subroutines, different from the noise-free simulation presented in Figure 3. As expected, the situation becomes worse when we gradually decrease the dephasing time. The effect of the noise is to reduce the algorithm’s accuracy, approximating the probability of the sought state to the other items of the unstructured list.

In order to obtain a landscape of the effect of the pure dephasing channel on the sought state, we plot the probability of the sought state as a function of the pure dephasing times ($T_{\phi }$) in Figure 7. As can be seen, as $T_{\phi }$ increases, the probability of the sought state approximates to a probability limit 95.86%, obtained for the noise-free simulation presented in Figure 3, highlighted in the dashed blue line. On the other hand, the performance of the search algorithm is negatively impacted by decreasing the dephasing time, rapidly decreasing the probability sought state. Therefore, we are convinced that these findings demonstrate that the existence of quantum noise in quantum processors reduces the efficiency of the algorithm, which can lead to inaccurate outcomes when simulating quantum circuits.

## 4. Conclusions

This paper presents the quantum search problem based on the famous Grover’s algorithm associated with the binary encoding of words into quantum states of the 4-qubit computational basis. We highlight the main conditions for developing projects and their implementation in dedicated hardware processors. Our findings are consistent with the theoretical predictions found in the aforementioned literature for the examples that were covered, and they demonstrate that AQASM and myQLM would be practical tools for both the implementation and analysis of quantum algorithms. Unlike other studies published in the literature, our work has contributed to the evaluation of the quality of Grover’s algorithm as a function of the pure dephasing quantum noise exclusively, made possible through the configurable features available in QLM. Moreover, we emulate a genuine quantum processor by configuring a dedicated quantum simulator to conform to the necessary architectural specifications for performing Grover’s algorithm. Furthermore, we demonstrate that the existence of quantum noise leads to a progressive decline in the quality of the outcomes produced by a quantum noisy simulation when compared to the result performed in a noise-free setup. Finally, we propose as a potential direction for future study the scaling up of our program to a greater number of qubits, which would result in an exponential increase in our database capacity.

## Figures and Tables

**Figure 1 entropy-26-00668-f001:**
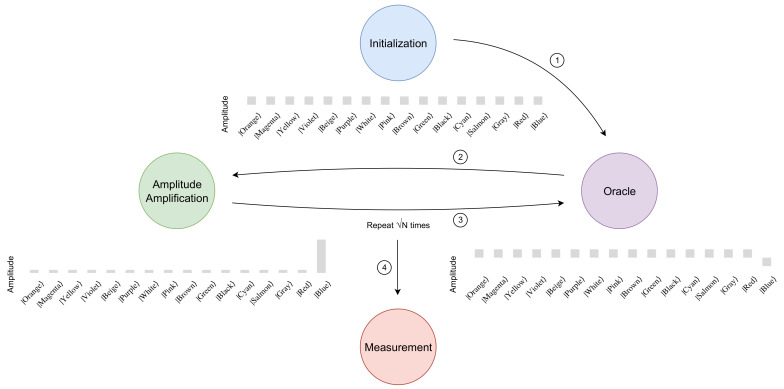
Sketch of the four steps of Grover’s algorithm along with the evolution of the probability amplitudes of each element of the 4-qubit computational basis.

**Figure 2 entropy-26-00668-f002:**
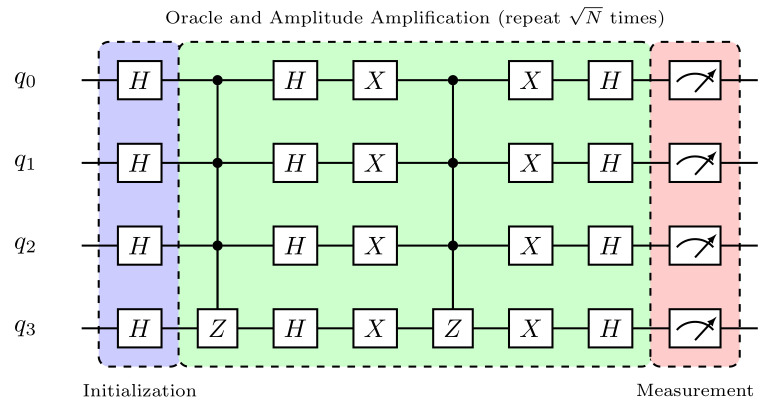
Grover’s circuit using myQLM (abbreviated).

**Figure 3 entropy-26-00668-f003:**
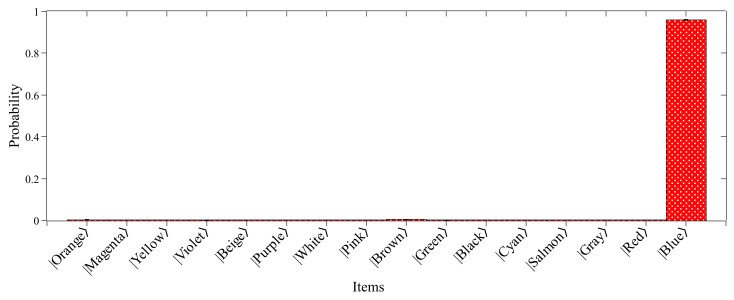
Probability distribution for the 16 items of the database. The searched item is found with 95.86% probability, while the other ones did not reach 0.5% probability individually.

**Figure 4 entropy-26-00668-f004:**
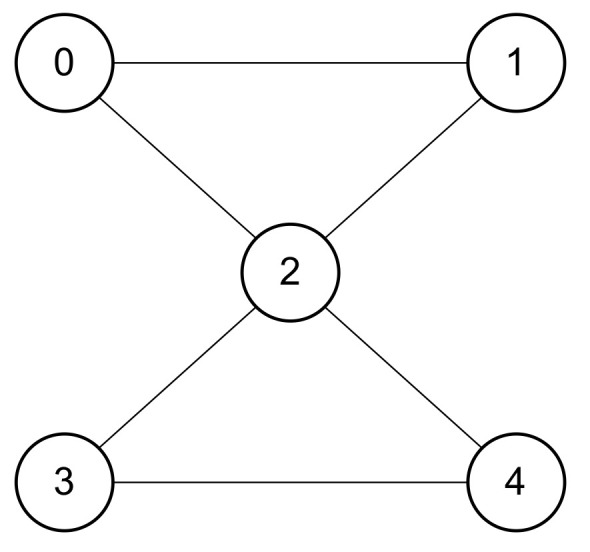
Sketch of the 5-qubit simulated topology. All qubits are coupled for simplicity since they can all be concurrently controlled or targeted by controlled quantum operations.

**Figure 5 entropy-26-00668-f005:**
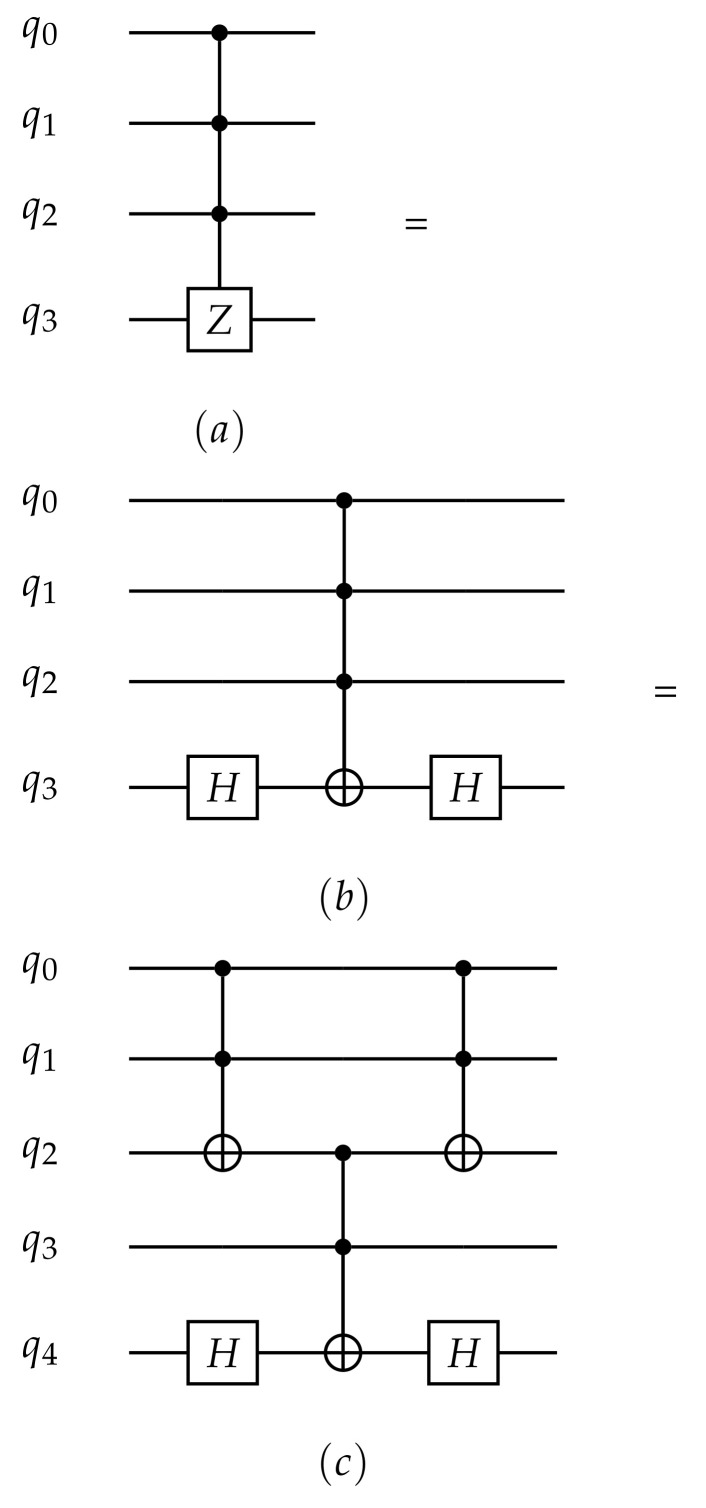
The decomposition of (**a**) multi-controlled Z gate (CCCZ) using (**b**) Hadamard and CCCNOT gates or (**c**) Hadamard and CCNOT gates.

**Figure 6 entropy-26-00668-f006:**
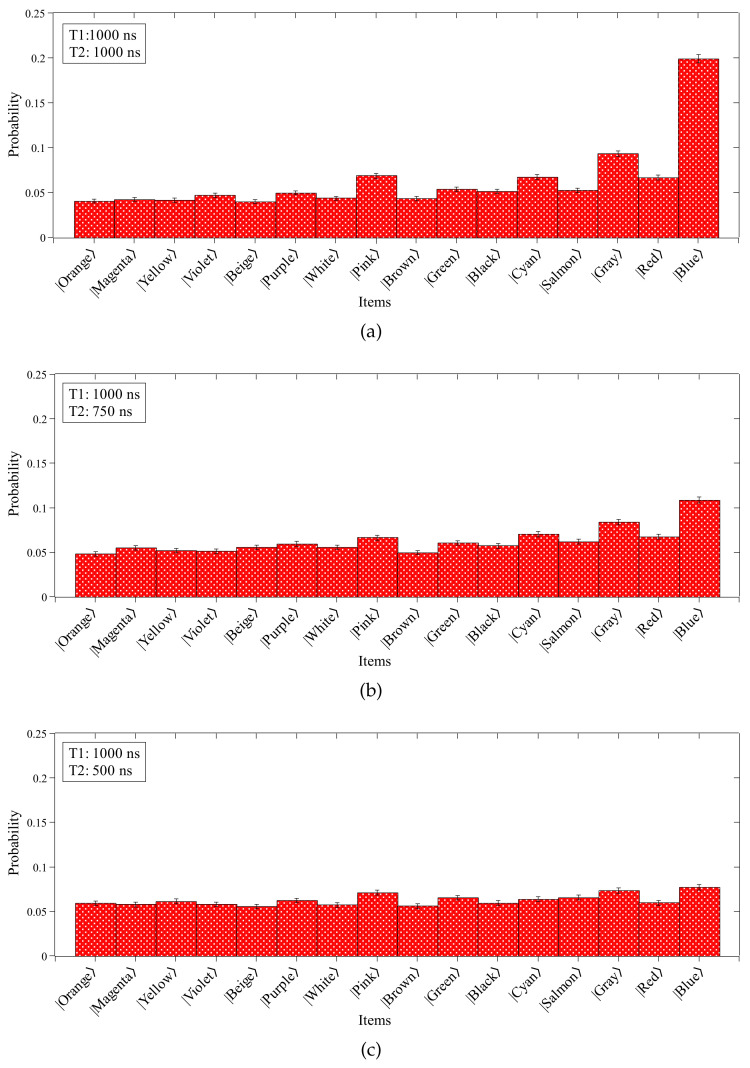
Simulation with noise using $T_{\phi }$ when $T_{1}$ is equal to 1000 ns and $T_{2}$ is equal to (**a**) 1000 ns, (**b**) 750 ns, and (**c**) 500 ns.

**Figure 7 entropy-26-00668-f007:**
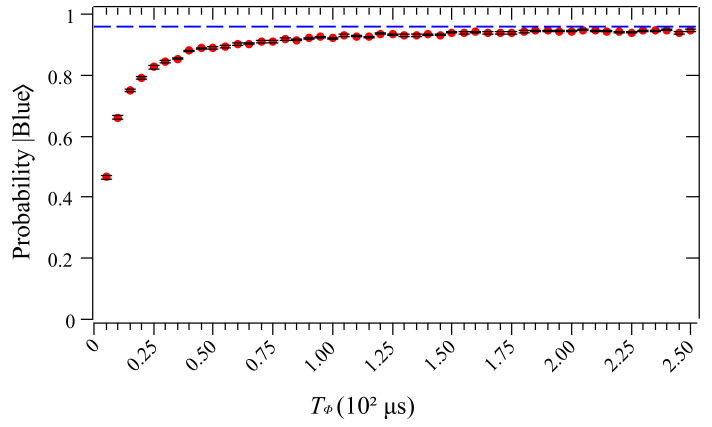
Probability distribution for item |Blue〉 as a function of pure dephasing times, $T_{\phi }$. The dashed blue line highlights the probability of the sought state (95.86%) obtained in the noise-free simulation presented in Figure 3.

## Data Availability

The data that support the findings of this study are openly available in Supplementary Information Grover Paper, a GitHub repository created by the authors [24].

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
