# Peer review of "Effect of Pure Dephasing Quantum Noise in the Quantum Search Algorithm Using Atos Quantum Assembly"

_entropy, 2024, doi:10.3390/e26080668_

Round 1

Reviewer 1 Report

Comments and Suggestions for Authors

The paper titled "Effect of Pure Dephasing Quantum Noise in the Quantum Search Algorithm Using Atos Quantum Assembly" presents the designs of 4-qubit Grover's algorithm using Quantum Assembly Language. The paper includes sufficient details and writes pedagogically. It is very friendly to beginners of quantum computation.

However, my major concern is that the paper is more about a tutorial than a research paper. Besides, the effects of noise on Grover's algorithm have been well studied. Another concern is the way of designing the oracle in the paper. It is simply a toy model. Designing the oracle requires the information of the target. This is not how the Grover's algorithm applied to practical problems. The authors should address this point in the paper, especially if the potential readers of the paper is the beginners. 

Reviewer 2 Report

Comments and Suggestions for Authors

Thanks for submitting to entropy. The biggest problem I have with this paper is low novelty.  The manuscript primarily covers basic concepts extensively, making it more suitable as a tutorial rather than a research paper. While it is a good educational material, its scientific contribution is unclear.

The presentation of the paper should be improved. Many unnecessary explanations and tables. For example, in Table1 there is no need to explicitly list binary and vector. Table2 should be moved to an appendix as the oracles do not need to be explicitly listed. 

Some other questions: 

1. How does your study and platform differ from existing quantum software platforms? Please highlight any unique features or improvements over current solutions.

2. The paper mentions hardware support of three qubit gates. Could you specify which hardware supports this feature? Is it a real device or simulator?

Comments on the Quality of English Language

The writing can be improved, particularly regarding the misuse of capital letters in several places.

Round 2

Reviewer 1 Report

Comments and Suggestions for Authors

The authors included more discussions on the novelty. The paper could be useful for researchers who are new to myQLM. Therefore, I recommend the publication to Entropy.